# Gender gap in journal submissions and peer review during the first wave of the COVID-19 pandemic. A study on 2329 Elsevier journals

Flaminio Squazzoni[1]*, Giangiacomo Bravo[2,3], Francisco Grimaldo[4], Daniel García-Costa[4], Mike Farjam[5], Bahar Mehmani[6]

**1** Department of Social and Political Sciences, University of Milan, Milan, Italy, **2** Centre for Data Intensive Sciences and Applications, Växjö, Sweden, **3** Department of Social Studies, Växjö, Sweden, **4** Department of Computer Science, University of Valencia, Burjassot, Spain, **5** European Studies, Centre for Languages and Literature, Lund University, Lund, Sweden, **6** STM Journals, Elsevier, Amsterdam, The Netherlands

* flaminio.squazzoni@unimi.it

**Data Availability Statement:** Data for findings replication are available at this link: https://doi.org/10.7910/DVN/S0T7Z5.

## Abstract

During the early months of the COVID-19 pandemic, there was an unusually high submission rate of scholarly articles. Given that most academics were forced to work from home, the competing demands for familial duties may have penalized the scientific productivity of women. To test this hypothesis, we looked at submitted manuscripts and peer review activities for all Elsevier journals between February and May 2018-2020, including data on over 5 million authors and referees. Results showed that during the first wave of the pandemic, women submitted proportionally fewer manuscripts than men. This deficit was especially pronounced among more junior cohorts of women academics. The rate of the peer-review invitation acceptance showed a less pronounced gender pattern with women taking on a greater service responsibility for journals, except for health & medicine, the field where the impact of COVID-19 research has been more prominent. Our findings suggest that the first wave of the pandemic has created potentially cumulative advantages for men.

## Introduction

The recent pandemic has spurred a flood of COVID-related research [1, 2]. Over 125,000 COVID-19–related papers were published in the first 10 months after the onset of the pandemic in 2020, of which more than 30,000 hosted by preprint servers [3]. The pandemic even increased the opportunities for publication in completely COVID-unrelated fields, such as ophthalmology [4]. Being a global, systemic challenge affecting nearly all the aspects of society, the pandemic has stimulated research on various health, economic, social and psychological factors [5], thus posing a challenge to journals called to handle an unprecedented volume of submissions at extraordinary speed [6].

However, from the onset of the pandemic, governments in many countries have enforced severe lockdown measures, requiring most academics to work from home. While academics are used to working at a distance and with flexible times, it is plausible that during the

**Funding:** FS is supported by a "Department of Excellence" grant from the Italian Ministry of Education, University and Research to the Department of Social and Political Sciences of the University of Milan and a Transition Grant from the University of Milan (PSR2015-17). FG and DG are partially supported by the Spanish Ministry of Science, Innovation and Universities (MCIU), the Spanish State Research Agency (AEI) and the European Regional Development Fund (ERDF) under project RTI2018-095820-B-I00. The funders had no role in study design, data collection and analysis, decision to publish, or preparation of the manuscript.

**Competing interests:** BM is employee of Elsevier and organised the data sharing process.

pandemic competing demands from homeschooling, family obligations and other caring duties have affected the productivity of women and men differently [7, 8]. Indeed, home-schooling and elderly care responsibilities due to COVID-19 lock down regulations have imposed a major shift in family schedules and routines, probably cementing even more traditional gender roles [9, 10]. It has long been known that women drop out more frequently from academia due to difficulties in reconciling work and family life [11–13]. Given that gender inequality in family and work are connected, it is reasonable to hypothesize that the pandemic could have deepened the pre-existing gender inequalities in both realms [14].

For instance, a study in the U.S. showed that women with young children have reduced their working hours four to five times more than fathers during the pandemic [15]. A survey on 4,535 principal investigators in scientific projects in Europe and the U.S. indicated that women academics, those in the 'bench sciences' and, especially, scientists with young children, have experienced a substantial decline in research time [16]. A recent perspective analysis suggested that the effect of the pandemic was worsened by the closure of laboratories and the interruption of most field and observational studies due to restrictions in response to the COVID-19 pandemic, as well as the freeze of intramural research accounts and extra-mural funding sources to support the medical mission [17].

From the early onset of the pandemic, the impression that women were submitting fewer manuscripts to journals was confirmed by two studies using PubMed database and data on preprints to estimate the gender rate of authors posting or publishing COVID-19 related papers during the pandemic [1, 18, 19]. A more recent study on the author byline of 42898 PubMed indexed life science articles found that the percentage of articles on which men versus women were first authors widened by 14 percentage points during the pandemic [20].

However, these findings are still controversial. While a study on American Journal of Public Health confirmed that submissions were higher from men [21], other studies in specific fields reported no trace of gender inequality in the proportion of submissions [22]. For instance, a study on the impact of the pandemic on six journals published by the British Ecological Society (BES) found that the proportion of submissions authored by women during the COVID period of 2020 did not change relative to the same period in 2019 [23].

Unfortunately, these studies either considered only preprints or publications, without access to data to examine submissions to journals, or lacked cross-journal data in various fields, thus limiting evidence to only specific cases. Understanding whether the COVID-19 race for publications has possibly disproportionately benefited men requires accessing full individual data from various journals in a comparable time frame before and during the pandemic, so as to estimate the effect of the pandemic on individual scholars. Although research on preprints is important to estimate the academic response to the greater demand for research during the pandemic [1, 19], looking at manuscript submissions and peer review activities for journals in different research areas before and during the pandemic is key to estimate gendered gaps in time and effort investment for research by academics more precisely.

To fill this gap, we have established a confidential agreement with Elsevier publishing to access manuscript and peer review metadata from all their journals. These included individual records of authors and referees in a fully comparable monthly time frame—i.e., February-May 2018–2020, during the first wave of the pandemic in Asia, Europe and America (see Methods Section). Note that focusing on the early months of the pandemic was instrumental to estimate gender inequalities as most countries enforced similar lockdown measures, which were eventually eased during summer.

Given that our data came from manuscript submission systems, we had to re-purpose them for research by adding gender guessing algorithms and mobility data from Google to control for residential data, and completing them with Scopus data to estimate scholar's age and

seniority. This allowed us to treat the pandemic as a 'quasi-experiment' and estimate its effect on academics' productivity at an individual level, by considering the seasonal rate of submissions in 2018–2020 in different research areas and residential countries. Unlike other studies, which looked at the gender proportion either of preprint or publication authors or submission authors [23], we examined the effect of the pandemic on each scholar active between 2018–2020 in these journal submission databases at the individual level. Furthermore, we included data on referees to understand whether women were penalized in their capacity to serve the community and influence the type of research performed during the pandemic.

## Materials and methods

### The dataset

Our dataset included complete information on manuscripts and reviews from 2329 Elsevier journals from January 2018 to May 2020 (see Table 1; S1 Table includes the total number of submissions in Feb-May 2018–2020). The sample included about 5 million academics listed as authors and/or referees. Data access required a confidential agreement to be signed on 12th May 2020 between Elsevier and each author of this study. The agreement was inspired by the PEERE protocol for data sharing and included anonymization, privacy, data management and security policies jointly determined by all partners [24].

  For the sake of our analysis, we concentrated on the first wave of the COVID-19 pandemic, i.e., from February to May 2020 (more precisely, weeks 6–22, 2020). This allowed us to cover the large part of the outbreak during the first half of 2020, including the effect of restrictions on mobility in China and Asia in Feb 2020 and in Europe and United States later. Furthermore, Google COVID-19 Community Mobility Report used the first five weeks of 2020 as reference so that mobility data were only available starting from week 6, 2020 (see https://www.google.com/covid19/mobility/; accessed on 30 June 2020). On the other hand, few countries had any lockdown measures in place during January 2020. To ensure full comparability across years, including seasonality issues, we decided to limit our observations to the corresponding months of 2018 and 2019.

  We used the e-mail (or the set of e-mails) associated to each user account in the underlying submission systems (i.e., Editorial Manager, Elsevier Editorial System and EVISE) to track academics across all journals and constructed an auto-generated anonymous unique identifier. We controlled for multiple e-mail addresses and this allowed us to circumvent the incompleteness of other alternative identifiers, which were either available only for a partial sub-sample of academics (e.g., ORCID) or not unique (e.g., ScopusID). Note that the same individuals may have been counted twice (or more) in the data reported in our analysis whenever submitting or reviewing to journals in different research areas.

Table 1. Overview of the main variables considered in the analysis by area of research.

|  | Health & Medicine | Life Sciences | Physical Sciences & Engineering | Social Sciences & Economics | Total |
|---|---|---|---|---|---|
| N. of journals | 885 | 416 | 767 | 261 | 2329 |
| Submissions (female) | 1005590 | 653729 | 991304 | 128798 | 2779421 |
| Submissions (male) | 1816621 | 1063178 | 2967128 | 271821 | 6118748 |
| Accepted reviews (female) | 133989 | 104065 | 194062 | 43319 | 475435 |
| Accepted reviews (male) | 359600 | 239918 | 888022 | 104621 | 1592161 |
| Declined reviews (female) | 233484 | 211185 | 336275 | 53476 | 834420 |
| Declined reviews (male) | 527723 | 441786 | 1338245 | 109969 | 2417723 |

To prevent de-anonymization of authors and referees, all submissions from countries with less than 20 authors/referees or with a number of authors that happens 5 times or less for the same journal were dropped from the dataset. This reduced our sample by 290082 submissions, i.e., about 6% of the observations. In addition to solving the privacy issue mentioned above, by removing observations from smaller countries we increased the robustness of the analysis, as the maximum likelihood estimation of random intercepts with few observations for each category may have caused convergence and over-fitting problems, thereby making it difficult to control possible statistical biases. Finally, these countries were also not covered by the Google COVID-19 Community Mobility Report and so should have been excluded in any case.

## Gender guessing

Our procedure for gender guessing was based on a two-step disambiguation algorithm inspired by previous research [25–28] and already validated on several datasets of academics' names [29]. First, we queried the Python package gender-guesser about the first names and countries of origin, if any. Gender-guesser allowed us to minimize gender bias and achieve the lowest mis-classification rate (less than 3% for Benchmark 1 in [29]). For names classified by gender-guesser as 'mostly_male', 'mostly_female', 'andy' (androgynous) or 'unknown' (name not found), we used GenderAPI (see https://gender-api.com/), which ensures that the level of mis-classification is around 5% (see Table 4 in [29]) and has the highest coverage on multiple name origins (see Table 5 in [29]). This procedure allowed us to guess the gender of 94.5% of academics in our sample, 45.1% coming from gender-guesser and 49.2% from GenderAPI. The remaining 5.5% of academics were assigned an unknown gender. Note that this level of gender guessing is consistent with the non-classification rate for names of academics in previous research [29]. Note also that while we were aware that any gender binary definition did not adequately represent non-binary identities, to the best of our knowledge, there was no better instrument to estimate gender for such a large pool of individuals.

We checked the robustness of the analysis to variations of the gender guessing algorithm by estimating further models using a more restrictive version of the algorithms, which kept the rate of miss-classified names resolved by GenderAPI under 5% and required a minimum of 62 samples with at least 57% accuracy (see S9 and S10 Tables where the percentage of academics without a guessed gender increased to 28.5%).

## Scholar's age

Scholars' age was estimated by using the number of years since their first record in the Scopus database. We followed a conservative rule and authors were identified by their Scopus IDs, e-mail addresses or the full name and country (in case a single profile was found). Authors without a profile in Scopus or not being uniquely identifiable were excluded from the analysis, whenever using age as a variable. Note that our aim here was not to estimate the age of each scholar precisely, which is impossible. We wanted to identify cohorts of academics to estimate the ones which most probably could have homeschooling and elderly care responsibilities. We assumed that first publications would correspond to the period in which academics were completing their MD or PhD period (i.e., estimated age around 25/30). We used this assumption to create the two cohorts mentioned in the text. The fact that our estimations could have mis-classified the actual age of scholars by some years (e.g., estimating someone being 40 instead of 43 years old in 2020) is irrelevant to the purpose of our study. Note that our classification could have underestimated the age of some authors who did not have any past formal training (e.g., a PhD title) and published their first paper only recently. While it is impossible to identify these cases in the database, our analysis using self-declared academic titles in Elsevier data

could be seen as a supplementary check on these cases, as they could be ideally listed themselves as Mr. and Ms. etc. and so being controlled for in our analysis.

Indeed, for a robustness check, we used the self-declared academic title and degree in the Elsevier dataset. Note that the use of the title "Dr." could be different in certain communities and perhaps not allowing to clearly identify someone with a PhD title. On the other hand, the title "Prof." could be used more rarely among academic faculty members working in hospitals. However, the size of the sample and the large coverage of academics from different countries and areas of research could have reduced the effect of this possible bias on our outcomes.

### COVID-19 related manuscripts

Elsevier data allowed us to distinguish COVID related and non-related manuscripts through an internal Boolean flag from the manuscript submission systems used by journals. A manuscript was considered COVID-19 related when the following condition was met by its keywords or abstract: ["covid-19" OR "covid 19" OR "covid19" OR "corona virus" OR "coronavirus" OR "corona-virus" OR "corona viruses" OR "coronaviruses" OR "corona-viruses" OR "orthocoronavirinae" OR "coronaviridae" OR "coronavirinae" OR "2019-ncov" OR "2019ncov" OR "2019 ncov" OR "hcov-19" OR "sars-cov" OR "sars cov" OR "severe acute respiratory syndrome" OR "sars-cov-2" OR "sars-cov2" OR "mers-cov" OR "mers cov" OR "middle east respiratory syndrome" OR "middle eastern respiratory syndrome" OR ("angiotensin-converting enzyme 2" AND "virus") OR ("ace2" AND "virus") OR "soluble ace2" OR ("angiotensin converting enzyme2" AND "virus") OR ("ards" AND "virus") OR "acute respiratory distress syndrome" OR ("sars" AND "virus") OR ("mers" AND "virus") OR ("wuhan" AND "virus")]. We used this taxonomy to track COVID-related manuscripts (i.e., manuscripts focusing on diseases caused by the the same family of viruses) before the start of the pandemic.

### Data analysis

All analyses were performed using the *R* platform [30]. The statistical analysis was performed exploiting the high-performance computing facility of the Linnaeus University Centre for Data Intensive Sciences and Applications. If not explicitly otherwise mentioned in the text, standard test to check the model assumptions (homogeneity, normality of random effect, etc.) were performed for all models.

## Results

The COVID-19 pandemic has caused an abnormal rate of journal submissions. Our data indicate that the number of manuscripts submitted to all Elsevier journals between February to May 2020 increased by 30% compared to the same period of the previous year (i.e., from 620,685 for February-May 2019 to 807,449 in 2020; the growth rate of submissions from 2018 to 2019 was 11%) (see S1 Table). Note that in health & medicine journals, this trend was even stronger with an increase of 63% (i.e., from 147,401 submissions from February-May 2019 to 240,587 in 2020). At the same time, the absolute numbers of accepted review invitations for all disciplines increased by 29%, from 1,847,256 in 2019 to 2,381,284 in 2020. In the case of health & medicine journals, the accepted invitations increased by 34% from 2019 to 2020 (i.e., 415,033 in 2019 against 554,895 in 2020) compared with an increase of 63% submissions.

Our analysis shows that while the number of manuscripts submitted to journals generally increased during the first wave of the pandemic, the number of manuscripts submitted by men was higher than those submitted by women (Fig 1A). The rate of accepted review invitations— i.e., the number of accepted invitations on the total number of invitations sent to potential

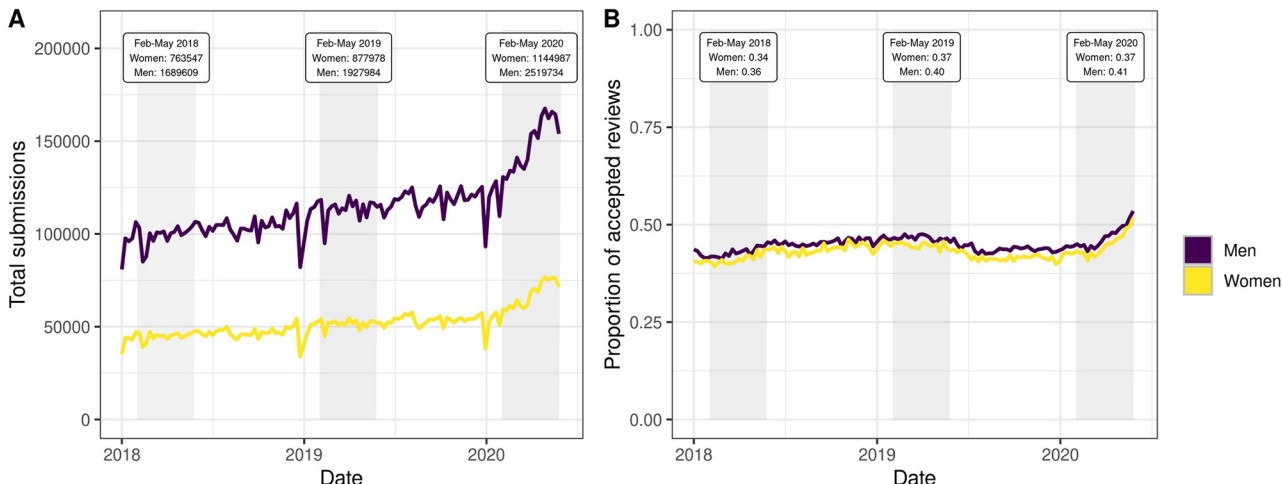

**Fig 1. Total submissions (A) and proportion of accepted reviews (B) per week across the whole period covered by the dataset.** The shaded areas indicate the February-May period of each year considered in the analysis. Note that in panel A co-authored submissions were reported multiple times depending on the number of co-authors. Each author or referee whose gender was not successfully guessed by our algorithm was excluded.

referees—has been more constant around an average of $\approx 40\%$ with women accepting slightly fewer invitations than men (Fig 1B).

In February-May 2018, 2019 and 2020, women submitted 2,779,421 manuscripts against 6,118,748 manuscripts submitted by men. Women agreed on performing 475,435 reviews while declining 834,420 invitations, with a proportion of 37% accepted invitations. Men accepted 1,592,161 review invitations while declining 2,417,723, with a similar acceptance proportion (40%) (see Table 1 for a summary of these descriptive statistics per research area).

## The effect of the pandemic on submissions

We calculated a submission difference index for each author ($\Delta_S$) as the number of new submissions in February-May 2020 minus the average number of submissions from the same author in the corresponding months of 2018 and 2019. We then estimated each scholar's age by using the number of years since their first record in the Scopus database. Given that students tend to complete their MD-PhD title when 25–30 years old [31, 32], we divided the sample in two age cohorts ($\leq$ or $> 20$ years after receiving their title), and hypothesized that more junior cohorts of women would be most likely affected by homeschooling and elderly care responsibilities.

Results showed that the overall increase of submissions in 2020 led most authors to $\Delta_S \geq 0$. However, when considering differences in age and areas of research, we found that the $\Delta_S$ of men increased more than that of women, especially those in the more junior cohort mentioned above (Fig 2). This would suggest that women had at least comparatively fewer opportunities for research during the first wave of the pandemic.

To check for the significance of these effects, we estimated a mixed effects model using authors' gender and age to predict $\Delta_S$ (Table 2). In order to control for the fact that authors were based in countries with different university systems and contagion-prevention policies, we included random effects for countries in the model as a geographical control. Results indicated a statistically significant negative effect for women in all areas of research (Table 2). In addition, we found a consistent positive interaction effect between gender and age, with more senior cohorts of women less penalized than younger scholars.

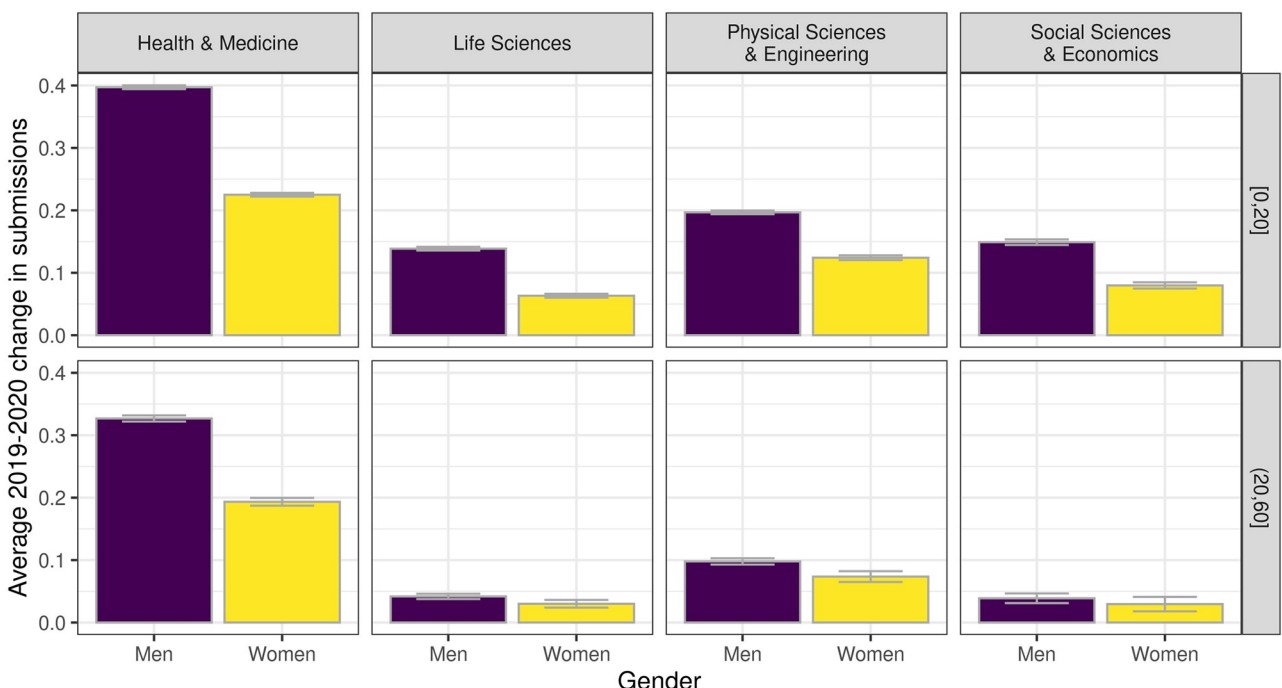

**Fig 2. Average change in submissions by research area and age, the latter variable including authors in the first cohort ($\leq$ 20 years from their first publication) in the first group with older authors in the second.** Bars represent standard errors.

As a robustness check of our age measurement, we estimated similar models using a measure of seniority based on the author's title (i.e., no title, Doctor or Professor) as recorded in Elsevier's database. Results confirmed previous findings. The deficit of women was pronounced especially in health & medicine, with a weakly significant effect in social sciences &

**Table 2. Mixed effects models predicting February-May 2020 changes in the number of submissions per area of research.**

|  | Health & Medicine | Life Sciences | Physical Sciences & Engineering | Social Sciences & Economics |
|---|---|---|---|---|
| Women | −0.164 | −0.078 | −0.097 | −0.077 |
|  | (0.007) | (0.007) | (0.008) | (0.011) |
|  | p < 0.001 | p < 0.001 | p < 0.001 | p < 0.001 |
| Age | −0.001 | −0.002 | −0.003 | −0.004 |
|  | (0.0002) | (0.0002) | (0.0002) | (0.0004) |
|  | p < 0.001 | p < 0.001 | p < 0.001 | p < 0.001 |
| Women×Age | 0.001 | 0.002 | 0.002 | 0.002 |
|  | (0.0004) | (0.0004) | (0.001) | (0.001) |
|  | p = 0.022 | p < 0.001 | p < 0.001 | p = 0.003 |
| Intercept | 0.329 | 0.138 | 0.209 | 0.185 |
|  | (0.020) | (0.015) | (0.018) | (0.014) |
|  | p <0.001 | p <0.001 | p <0.001 | p <0.001 |
| Observations | 706126 | 480240 | 856454 | 152348 |
| Log Likelihood | −1369462 | −818103 | −1816437 | −245405 |

The baseline is represented by the average of corresponding months in 2018 and 2019. Random intercepts included for countries.

**Table 3. Mixed effects models predicting February-May 2020 changes in the number of submissions per area of research.**

| | Health & Medicine | Life Sciences | Physical Sciences & Engineering | Social Sciences & Economics |
|---|---|---|---|---|
| Women | −0.080 | −0.0003 | −0.016 | −0.028 |
| | (0.008) | (0.008) | (0.008) | (0.012) |
| | p <0.001 | p = 0.973 | p = 0.055 | p = 0.016 |
| Doctor | 0.089 | 0.009 | 0.058 | 0.026 |
| | (0.007) | (0.006) | (0.006) | (0.009) |
| | p <0.001 | p = 0.126 | p <0.001 | p = 0.004 |
| Professor | 0.163 | 0.017 | 0.091 | 0.013 |
| | (0.008) | (0.007) | (0.007) | (0.010) |
| | p <0.001 | p = 0.020 | p <0.001 | p = 0.217 |
| Women×Doctor | −0.027 | −0.021 | −0.016 | 0.005 |
| | (0.009) | (0.009) | (0.010) | (0.014) |
| | p = 0.005 | p = 0.022 | p = 0.115 | p = 0.708 |
| Women×Professor | −0.065 | −0.010 | 0.001 | 0.011 |
| | (0.013) | (0.012) | (0.014) | (0.018) |
| | p <0.001 | p = 0.405 | p = 0.971 | p = 0.552 |
| Intercept | 0.314 | 0.172 | 0.182 | 0.188 |
| | (0.019) | (0.016) | (0.017) | (0.015) |
| | p <0.001 | p <0.001 | p <0.001 | p <0.001 |
| Observations | 847892 | 571051 | 1026221 | 195157 |
| Log Likelihood | −1635935 | −984773 | −2180291 | −314846 |

The baseline is represented by the average of corresponding months in 2018 and 2019. Models include as predictor the title of each author with no title used as reference category. Random intercepts included for countries.

economics and physical sciences (Table 3). Note that we considered any value of $0.05 < p < 0.005$ as being only weakly significant [33].

In order to test the hypothesis that these gender and age differences were a side-effect of the different anti-contagious measures adopted by various countries, we included in the model a proxy of how lockdown and social distancing measures, such as the closure of schools, could have affected academics in different countries. Following recent geographical research on the effect of contagion-prevention measures [34–36], we used Google's COVID-19 Community Mobility Report (see details in the supplementary materials), which tracks the amount of time spent by mobile-phone users in different places, including residential areas. Mobility reports are available at the country level (in some cases even at sub-national level) and are summarised in an index that calculates the different time rates spent by individuals in residential areas in a given day compared to the median value of January 2020.

We calculated the average values of the February-May 2020 period of the residential area index per country (see the map in S1 Fig) to control for the exposure of each scholar to the same mobility restrictions and lockdown measures. Unfortunately, certain countries (e.g., China and Iran) were not included in the mobility reports and so our analysis was performed on a restricted sample of academics (this caused a reduction of our observations from 16% to 32% depending on the area of research; see Table 4).

Results indicated a negative interaction between gender and time in residential areas when considering authors submitting manuscripts to health & medicine, and physical science & engineering journals. In addition, we found a significant or weakly significant and negative pure effect of gender in all areas of research (Table 4).

**Table 4. Mixed effects models predicting February-May 2020 changes in the number of submissions per area of research.**

|  | Health & Medicine | Life Sciences | Physical Sciences & Engineering | Social Sciences & Economics |
|---|---|---|---|---|
| Women | −0.056 | −0.058 | −0.052 | −0.053 |
|  | (0.016) | (0.016) | (0.020) | (0.026) |
|  | $p = 0.001$ | $p < 0.001$ | $p = 0.010$ | $p = 0.041$ |
| Age | −0.002 | −0.002 | −0.003 | −0.003 |
|  | (0.0002) | (0.0002) | (0.0003) | (0.0004) |
|  | $p < 0.001$ | $p < 0.001$ | $p < 0.001$ | $p = <0.001$ |
| Residential | 0.018 | 0.005 | 0.003 | 0.003 |
|  | (0.004) | (0.003) | (0.004) | (0.003) |
|  | $p = <0.001$ | $p = 0.147$ | $p = 0.522$ | $p = 0.279$ |
| Women×Age | 0.002 | 0.002 | 0.002 | 0.002 |
|  | (0.0004) | (0.0004) | (0.001) | (0.001) |
|  | $p < 0.001$ | $p < 0.001$ | $p < 0.001$ | $p = 0.008$ |
| Women×Residential | −0.010 | −0.002 | −0.004 | −0.001 |
|  | (0.001) | (0.001) | (0.001) | (0.002) |
|  | $p < 0.001$ | $p = 0.074$ | $p = 0.005$ | $p = 0.490$ |
| Intercept | 0.096 | 0.090 | 0.181 | 0.138 |
|  | (0.052) | (0.044) | (0.053) | (0.041) |
|  | $p = 0.067$ | $p = 0.041$ | $p = 0.001$ | $p = 0.001$ |
| Observations | 587184 | 356988 | 577890 | 127263 |
| Log Likelihood | −1098897.000 | −581594.400 | −1175353.000 | −200281.500 |

The baseline is represented by the average of corresponding months in 2018 and 2019. Models include time in residential areas from Google's COVID-19 Community Mobility Report (see https://www.google.com/covid19/mobility/; accessed on 30 June 2020). Random intercepts included for countries.

We then concentrated on 'COVID-related' manuscripts, i.e., manuscripts focusing on diseases caused by viruses of the *Coronaviridae* family (see Materials and methods). By using keywords similarity and internal classifications from Elsevier, we reconstructed the time trends of 'COVID-related' manuscripts submitted by academics to all Elsevier journals in the same period in 2018–2020, e.g., research on SARS-CoV-1. This also allowed us to focus on whether women doing research more directly relevant to COVID-19 were penalised during the pandemic.

Results confirmed that women submitted fewer COVID-19 related manuscripts in 2020 in health & medicine journals (Table 5). Note that we found non-significant or weakly-significant coefficients in other areas of research because of the relatively lower number of COVID-19 related manuscripts submitted to these journals (see S3 Table).

**Table 5. Mixed effects models predicting February-May 2020 changes in the number of submissions of COVID-related manuscripts in health & medicine journals.**

|  | Estimate | Std. Error | t | p |
|---|---|---|---|---|
| Intercept | 1.421 | 0.037 | 38.007 | <0.001 |
| Women | -0.133 | 0.027 | -4.967 | <0.001 |
| Age | 0.003 | 0.001 | 3.304 | 0.001 |
| Women×Age | -0.001 | 0.002 | -0.857 | 0.392 |
| Observations | 51916 |  |  |  |
| Log Likelihood | -99295 |  |  |  |

Random intercepts included for countries.

We also performed similar analyses on other specific subsets of manuscripts or journals. For instance, we considered the type of submissions by concentrating only on manuscripts indexed as "research papers" (see S4 Table), on submissions to first quartile (Q1) journals in the 2020 Journal Citation Reports (see S2 and S5 Tables), on the gender of first authors (see S6 Table), and on manuscripts with only one author (see S7 Table). Results of these analyses were fully consistent with our main finding: women academics submitted fewer manuscripts, both research and non-research manuscripts (e.g., commentaries), either in Q1 or in journals with a lower impact factor, and also fewer manuscripts as first authors. This effect was significant for more junior cohorts of women except for manuscripts submitted as first authors, where women were penalized regardless of age. In the case of single-authored manuscripts, the intersection of gender and age was significant only for submissions to health & medicine journals, with gender having a negative effect on submissions in all research areas except in physical science journals.

### The effect of the pandemic on academics' commitment to peer review

In order to measure the gender effect of the COVID-19 pandemic on academics' commitment to peer review, we calculated the proportion of review invitations accepted $\Delta_R$ for each invited referee in February-May 2020 compared to the corresponding period in 2019. This proportion excluded individuals who did not receive any invitations in February-May 2019 and 2020. To minimize missing values, we excluded the 2018 sample and restricted our analysis to February-May 2019 and 2020.

Besides an overall decline in the number of accepted invitations per individual, our results showed that the pandemic generally did not determine considerable gender differences by either research areas or scholar's age, the sole exceptions being peer review in health & medicine and physical science journals and the case of more senior referees (see Fig 3).

We then estimated two mixed-effect models per area of research, controlling for the time spent in residential areas. Results confirmed that the relative decline in $\Delta_R$ was more pronounced only for women in health & medicine (Table 6), although even in this case, we did not find any significant interaction with the time spent in residential areas (see S8 Table).

## Discussion

The COVID-19 pandemic has generated unforeseen opportunities for research as a collective response of the academic community to the pandemic [1, 8]. While this will continue over the following months due to various factors, including the challenge of possible mutations of the virus, and the global, societal implications of the pandemic, the exceptional lockdown and social distancing measures introduced since the first wave of the pandemic in early 2020 could have created inequalities in this academic race due to the competing demands for homeschooling and other family care duties [9, 16, 37].

Our complete data on all Elsevier journals indicate that women submitted fewer manuscripts than men during the first wave of the pandemic in early 2020. This has been especially prominent in the research area where the academic production has been higher during the pandemic, i.e., health & medicine. This suggests that the pandemic could have exacerbated existing inequalities by imposing additional obstacles in terms of time and effort investment by women just as the demand for research was growing unprecedentedly.

Our findings suggest that more junior cohorts of women academics were penalized the most. If we consider our control for residential mobility, this could be possibly explained by a major shift in family schedules and routines caused by the pandemic due to interference of homeschooling and more intense family duties, which could have seen these cohorts of

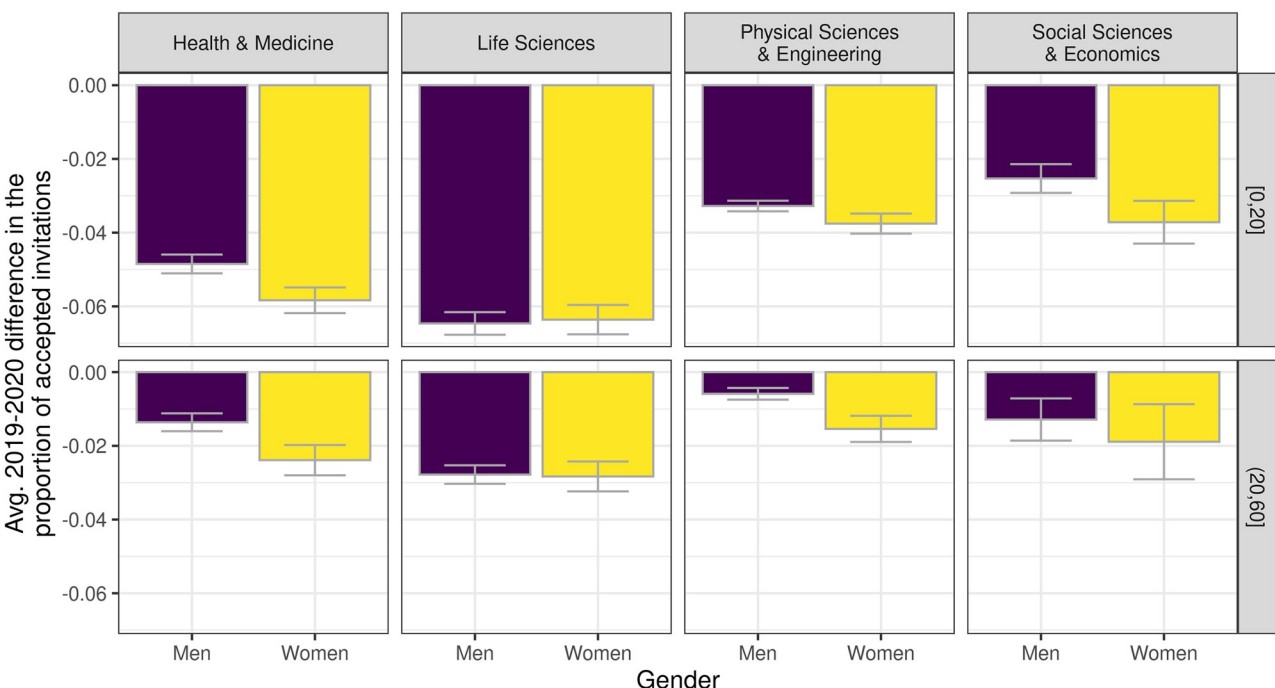

**Fig 3. Average difference in the proportion of accepted invitations by areas of research and age, the latter variable including authors in the more junior cohort (≤ from their first publication) in the first group with more senior authors in the second.** Bars represent standard errors.

women on the front-line [15]. Note that these cohorts would probably include women without permanent academic positions, competing for tenure, promotion and grants.

While pressures on peer review are higher in this period, requiring also special arrangements by many journals—e.g., special fast-tracks—, our findings suggest a general decline of accepted invitations but without pronounced gender effects. On the one hand, our findings

**Table 6. Mixed effects models predicting February-May 2020 changes in the proportion of accepted review invitations per area of research.**

|  | Health & Medicine | Life Sciences | Physical Sciences & Engineering | Social Sciences & Economics |
|---|---|---|---|---|
| Women | −0.016 | −0.001 | −0.005 | −0.016 |
|  | (0.007) | (0.008) | (0.005) | (0.012) |
|  | p = 0.025 | p = 0.914 | p = 0.319 | p = 0.173 |
| Age | 0.002 | 0.002 | 0.001 | 0.001 |
|  | (0.0002) | (0.0002) | (0.0001) | (0.0003) |
|  | p < 0.001 | p < 0.001 | p < 0.001 | p = 0.005 |
| Women×Age | 0.0003 | 0.0001 | 0.00002 | 0.0005 |
|  | (0.0003) | (0.0003) | (0.0002) | (0.001) |
|  | p = 0.277 | p = 0.648 | p = 0.944 | p = 0.500 |
| Intercept | −0.068 | −0.086 | −0.052 | −0.037 |
|  | (0.005) | (0.005) | (0.003) | (0.007) |
|  | p < 0.001 | p < 0.001 | p < 0.001 | p < 0.001 |
| Observations | 90902 | 78491 | 206426 | 29287 |
| Log Likelihood | −55689 | −49400 | −123204 | −19882 |

The baseline is represented by the average of the corresponding months in 2019. Random intercepts included for countries.

indicate that women have taken on a greater service responsibility for journals and the community as referees at least comparatively comparable to men. At the same time, men have submitted more manuscripts, thus benefiting from the involvement of women as referees. On the other hand, women were less involved in peer review for health & medicine journals, the field where the impact of COVID-19 research has been more prominent. This would suggest that they were less capable of influencing the type of research that was published. This raises concern over the quality of peer review under increasing editorial pressures during the pandemic, which would require an entire follow up study [38].

This said, our study has certain limitations. Though we achieved an observation scale never achieved previously in this type of research, our sample was limited to only Elsevier journals. However, Elsevier does have one of the largest journal portfolios of all publishers, sufficiently covers all areas of research, and represents a balanced proportion of journals across research areas (see S2 Table). While a desirable extension would be to expand this analysis by including journals from other publishers, we must acknowledge that creating a common database with full data on submissions from different publishers is at the moment impossible due to lack of a data sharing infrastructure solving legal and technical obstacles and creating opportunities for cooperation [39].

Finally, as mentioned above, unfortunately Google mobility data were not available in certain countries and regions, e.g., China and Iran. Therefore, we could not include our lockdown proxy (extra time spent in residential areas) for all observations in the sample. This suggests to consider our models including mobility data more as a robustness check for our analysis. Note, however, that any other possible measurements of actual lockdown of our sampled academics, such as country-based dates when these measures were introduced, were intrinsically biased because individuals could anticipate these announcements by staying at home before their introduction and/or even after the specific dates when restrictions are removed.

Given that many submissions during the pandemic will eventually turn into publications and citations, and considering the importance of these latter for academic career and prestige, it is probable that the first wave of the pandemic that we have examined here could be seen as the *genealogy* of gender disparities that will have important short- and longer-term effects. Pandemics have always exacerbated existing inequalities [40]. Indeed, those who have already benefited from this COVID-19 research race may have better chances in the near future to receive prestigious grants and obtain tenures and promotion in prestigious institutions. Previous research on peer review and editorial processes at journals has shown that gender inequalities in the rate of submissions to journals is key to determine inequality of publications and recognition [41].

In conclusion, it is important that funding agencies and hiring and promotion committees at national and international levels reconsider their policies in these exceptional times. While voluntary disclosure of gender or gender quotas during journal submissions could lead to further biases [19], flagging, carefully pondering or even disregarding COVID-19 related publications and citations from applicants' assessment could be considered. Following the example of the Canadian Institutes of Health Research (CIHR), extending deadlines and supporting COVIGiven that the use of bibliometric indicators to assess applicants for funds and academic positions has been strongly criticized even in normal times [42], one of the most important lessons from the pandemic could be to follow multi-dimensional criteria in any academic assessment. This could include a COVID-19 impact statement where any candidate is required to explain the opportunities and constraints faced during the pandemic [43].

At the same time, improving career enhancement and retention by appropriate institutional interventions, such as promoting a more diverse, inclusive, and equitable working environment and embracing a family-friendly leadership policy in the management of labs and

institutes, could help moderate the distortions caused by the pandemic [44]. These interventions could transform the pandemic in an unprecedented opportunity to reset certain established practices and reconsider how funders, institutes and universities could offer better support to academics who are more vulnerable to the effect of global crisis [45].

In this context, journals and publishers should increase their usual effort in internal assessment and monitoring with a special focus on the consequences of the pandemic on research [23, 46]. This study has paved the way for large-scale collaboration initiatives on data sharing between publishers and the scientific community [39] and could be used as a template to map the evolution of the pandemic science.

## Supporting information

**S1 Fig. Average increase in the time spent in residential areas by country.** The change was calculated as different rate from the baseline given by median value during the first five weeks of 2020. Data from Google COVID-19 Community Mobility Report (see https://www.google.com/covid19/mobility/; accessed on 30 June 2020). White areas indicate missing data.
(TIF)

**S1 Table. Total number of new submissions, review invitations, and accepted invitations per area of research in February-May 2020 and corresponding months of 2018 and 2019.** Note that data reported here differ from those in Table 1 because: (i) several authors could have submitted the same manuscript to different journals, which was only counted once here; and (ii) submissions and reviews from academics whose gender was not guessed by our algorithm were included here but not in Table 1.
(PDF)

**S2 Table. Proportion (%) of journals included in each quartile of the impact factor distribution by area of research.** The quartiles were calculated using Journal Citation Reportsby Clarivate Analytics.
(PDF)

**S3 Table. Mixed effects models predicting February-May 2020 changes in the number of submissions of Covid-related manuscripts per area of research.** Random intercepts included for countries.
(PDF)

**S4 Table. Mixed effects models predicting February-May 2020 changes in the number of submissions of research papers.** The baseline is represented by the average of corresponding months in 2018 and 2019. Random intercepts included for countries.
(PDF)

**S5 Table. Mixed effects models predicting February-May 2020 changes in the number of submissions of manuscripts submitted to Q1 journals.** The baseline is represented by the average of corresponding months in 2018 and 2019. Random intercepts included for countries.
(PDF)

**S6 Table. Mixed effects models predicting February-May 2020 changes in the number of submissions by first authors.** The baseline is represented by the average of corresponding months in 2018 and 2019. Random intercepts included for countries.
(PDF)

**S7 Table. Mixed effects models predicting February-May 2020 changes in the number of submissions by solo authors.** The baseline is represented by the average of corresponding months in 2018 and 2019. Random intercepts included for countries.
(PDF)

**S8 Table. Mixed effects models predicting February-May 2020 changes in the proportion of accepted review invitations per area of research.** The baseline is represented by the average of the corresponding months in 2019. Models included time in residential areas from Google's COVID-19 Community Mobility Report (see https://www.google.com/covid19/mobility/; accessed on 30 June 2020). Random intercepts included for countries.
(PDF)

**S9 Table. Mixed effects models predicting February-May 2020 changes in the number of submissions per area of research area.** The baseline is represented by the average of corresponding months in 2018 and 2019. Random intercepts included for countries. Gender data based on the stricter version of the gender guessing algorithm.
(PDF)

**S10 Table. Mixed effects models predicting February-May 2020 changes in the proportion of accepted review invitations per area of research.** The baseline is represented by the average of the corresponding months in 2019. Random intercepts included for countries. Gender data based on the stricter version of the gender guessing algorithm.
(PDF)

## Acknowledgments

We gratefully acknowledge the support on data extraction from the IT staff of Elsevier, specifically Ramsundhar Baskaravelu and his team. We also thank Dave Santucci from Elsevier Scopus API team and Kristy James from Elsevier International Center for Study of Research (ICSR) for their support on data enrichment about authors and reviewers. The statistical analysis was performed exploiting the high-performance computing facilities of the Linnaeus University Centre for Data Intensive Sciences and Applications.

## Author Contributions

**Conceptualization:** Flaminio Squazzoni, Giangiacomo Bravo, Francisco Grimaldo, Daniel García-Costa, Mike Farjam, Bahar Mehmani.

**Data curation:** Francisco Grimaldo, Daniel García-Costa, Bahar Mehmani.

**Formal analysis:** Giangiacomo Bravo, Francisco Grimaldo, Daniel García-Costa, Mike Farjam.

**Investigation:** Flaminio Squazzoni.

**Methodology:** Flaminio Squazzoni, Giangiacomo Bravo, Francisco Grimaldo, Daniel García-Costa.

**Project administration:** Flaminio Squazzoni, Bahar Mehmani.

**Supervision:** Flaminio Squazzoni, Francisco Grimaldo.

**Validation:** Francisco Grimaldo, Daniel García-Costa.

**Visualization:** Francisco Grimaldo, Daniel García-Costa.

**Writing – original draft:** Flaminio Squazzoni, Giangiacomo Bravo, Francisco Grimaldo, Daniel García-Costa, Mike Farjam, Bahar Mehmani.

**Writing – review & editing:** Flaminio Squazzoni, Giangiacomo Bravo, Francisco Grimaldo, Daniel García-Costa, Mike Farjam, Bahar Mehmani.

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
