## [Decision Letter · Decision Letter 0]

25 Jun 2021

PONE-D-21-15330

Only Second-Class Tickets for Women in the COVID-19 Race. A Study on Manuscript Submissions and Reviews in 2329 Elsevier Journals

PLOS ONE

Dear Dr. Squazzoni,

Thank you for submitting your manuscript to PLOS ONE. After careful consideration, we feel that it has merit but does not fully meet PLOS ONE’s publication criteria as it currently stands. Therefore, we invite you to submit a revised version of the manuscript that addresses the points raised during the review process.

The two reviewers are in agreement in judging positively your paper. Hence, I would only recommend to consider carefully all the suggestions of the two reviewers before the resubmission. 

We look forward to receiving your revised manuscript.

Kind regards,

Alberto Baccini, Ph.D.

Academic Editor

PLOS ONE

Journal Requirements:

2.  Please consider changing the title so as to meet our title format requirement (https://journals.plos.org/plosone/s/submission-guidelines). In particular, the title should be "Specific, descriptive, concise, and comprehensible to readers outside the field" and in this case it is not informative and specific about your study's scope and methodology (in particular the first part of the title).

3. Please improve statistical reporting and refer to p-values as "p<.001" instead of "p=.000". Our statistical reporting guidelines are available at https://journals.plos.org/plosone/s/submission-guidelines#loc-statistical-reporting

4. The Materials and Methods section should provide enough detail to allow suitably skilled investigators to fully replicate your study (https://journals.plos.org/plosone/s/submission-guidelines#loc-parts-of-a-submission). Please consider moving methodological details of your study from the Supporting Information to the body of the manuscript.

5. In the ethics statement in the submission form and the methods section, please confirm whether authors' and reviewers' last names were removed in the dataset you used in your study before you accessed it.

6. Thank you for stating the following in the Financial Support Section of your manuscript:

"FS is supported by a "Department of Excellence" grant from the Italian

Ministry of Education, University and Research to the Department of Social and Political

Sciences of the University of Milan. FG and DG are partially supported by the Spanish Ministry

of Science, Innovation and Universities (MCIU), the Spanish State Research Agency (AEI) and

the European Regional Development Fund (ERDF) under project RTI2018-095820-B-I00."

7. We note that you have stated that you will provide repository information for your data at acceptance. Should your manuscript be accepted for publication, we will hold it until you provide the relevant accession numbers or DOIs necessary to access your data. If you wish to make changes to your Data Availability statement, please describe these changes in your cover letter and we will update your Data Availability statement to reflect the information you provide.

8. We note that Figure S1 in your submission contain map images which may be copyrighted. All PLOS content is published under the Creative Commons Attribution License (CC BY 4.0), which means that the manuscript, images, and Supporting Information files will be freely available online, and any third party is permitted to access, download, copy, distribute, and use these materials in any way, even commercially, with proper attribution. For these reasons, we cannot publish previously copyrighted maps or satellite images created using proprietary data, such as Google software (Google Maps, Street View, and Earth). For more information, see our copyright guidelines: http://journals.plos.org/plosone/s/licenses-and-copyright.

8.1.    You may seek permission from the original copyright holder of Figure S1 to publish the content specifically under the CC BY 4.0 license. 

8.2.    If you are unable to obtain permission from the original copyright holder to publish these figures under the CC BY 4.0 license or if the copyright holder’s requirements are incompatible with the CC BY 4.0 license, please either i) remove the figure or ii) supply a replacement figure that complies with the CC BY 4.0 license. Please check copyright information on all replacement figures and update the figure caption with source information. If applicable, please specify in the figure caption text when a figure is similar but not identical to the original image and is therefore for illustrative purposes only.

Reviewers' comments:

Reviewer's Responses to Questions

**Comments to the Author**

1. Is the manuscript technically sound, and do the data support the conclusions?

Reviewer #1: Yes

Reviewer #2: Yes

2. Has the statistical analysis been performed appropriately and rigorously? 

Reviewer #1: Yes

Reviewer #2: Yes

3. Have the authors made all data underlying the findings in their manuscript fully available?

Reviewer #1: Yes

Reviewer #2: Yes

4. Is the manuscript presented in an intelligible fashion and written in standard English?

Reviewer #1: Yes

Reviewer #2: Yes

5. Review Comments to the Author

Reviewer #1: This manuscript investigates the gap between female and male scientists in terms of their manuscript submissions and review activities under the current COVID-19 pandemic, using data from 2329 Elsevier journals. The study is innovative because it used rarely seen large-scale data for both manuscript submissions and peer reviews.

Additional comments:

(1) The manuscript does not have a literature review section. Some related work was mentioned in the introduction section of the manuscript, but providing additional literature survey will allow the readers to understand the topic much better.

(2) The overall data used in the study was from Feb-May 2018 to 2020. For comparisons between women and men’s submission, Feb-May 2019 and 2020 was used. For most countries, except China, social distancing measures and school closures did not happen until the middle of March (e.g. U.S.). Additionally, considering that for most of the manuscripts submitted in or before May, much of the work would have happened way before the world “shut down” due to the pandemic in most of the countries. Therefore, the time span used in the analysis is a little problematic. It will be useful to know what was the annual increase rate for submissions and reviews in Elsevier journals before the pandemic to rule out the possibility that the 30% increase in submissions between Feb-May 2019 and 2020 are in fact not due to normal annual increase.

(3) For results in Table S4, the text in the manuscript says the dependent variable is the change in submissions for individual academics, while the supplemental materials indicate that the dependent variable is the number of submissions. The inconsistency needs to be adjusted.

(4) On page 6: “in addition, we ….with more senior cohorts of women less penalized than younger scholars.” I am having hard time concluding this from the results in Table S4. I think it will be better if authors can explain a little bit more here.

(5) What is the percentage of “not title”, “doctor”, and “professor” category in the database? It will be useful to know in order to access the robustness of the analysis

(6) In all the analysis, how were those researchers entered “publishing” during the pandemic handled, considering they did not have publication histories prior to this? For the analysis on COVID-19 related submissions, this will be a bigger issue since there are a lot of people who did not publish on coronavirus related topics at all before they did so for COVID-19, which is particularly true of those non-biomedical COVID-19 publications.

(7) Based on the regression analysis results, the increase of submissions from female authors were smaller than that from male authors.

Reviewer #2: I find the paper very interesting, and I think the penalisation of (especially young) women is indisputable in all areas, including academia. A supporting analysis definitely deserves attention.

Concerning the methodology, I must say that I am not an expert in linear mixed models, but I have some general comments.

The first critical point concerns the data used. In a first analysis, to estimate the gender effects of the pandemic on submissions, the authors calculate the submission difference index for each author (dependent variables) as the difference between new submissions February-May 2020 minus the average number of submissions for the same months and for the same author in the periods 2018 and 2019. I wonder how plausible it is for the same author to submit an article in the same months in the previous two years. I think the presence of unobservable variables is not taken into much consideration. Moreover, February-May 2020 seems to me to be a very limited period, but above all too "early", to see the effects on scientific production, especially in the West. Only in Italy, one of the first countries to be affected, have there been closures since March. In this regard, the authors control for different "geographical" measures of prevention by including random effects for countries. It is very interesting when they consider the scientific production on Covid-19 topics only, which surely has more concise working times, given the need of researchers to submit quickly. In that case, considering the time span February (maybe a bit early) - May seems to me more plausible.

Furthermore, although it is clear that the focus of the paper is certainly not methodological, I think that a section (also in the supplementary materials) in which to motivate why linear mixed models have been considered and a minimum of formalisation is necessary. In addition, there is no test to verify the assumptions of the model (e.g. homogeneity, normality of error term, normality of random effect). Tests have probably been done, but this should at least be mentioned.

6. PLOS authors have the option to publish the peer review history of their article (what does this mean?). If published, this will include your full peer review and any attached files.

Reviewer #1: No

Reviewer #2: No

---

## [Author Response · Author response to Decision Letter 0]

21 Jul 2021

Reviewers' comments:

Reviewer's Responses to Questions

Comments to the Author

1. Is the manuscript technically sound, and do the data support the conclusions?

Reviewer #1: Yes

Reviewer #2: Yes

2. Has the statistical analysis been performed appropriately and rigorously?

Reviewer #1: Yes

Reviewer #2: Yes

3. Have the authors made all data underlying the findings in their manuscript fully available?

Reviewer #1: Yes

Reviewer #2: Yes

4. Is the manuscript presented in an intelligible fashion and written in standard English?

Reviewer #1: Yes

Reviewer #2: Yes

5. Review Comments to the Author

Reviewer #1: This manuscript investigates the gap between female and male scientists in terms of their manuscript submissions and review activities under the current COVID-19 pandemic, using data from 2329 Elsevier journals. The study is innovative because it used rarely seen large-scale data for both manuscript submissions and peer reviews.

Additional comments:

(1) The manuscript does not have a literature review section. Some related work was mentioned in the introduction section of the manuscript, but providing additional literature survey will allow the readers to understand the topic much better.

We have improved the background section by adding further (and more recent) references and elaborating more on the context and purposes of the study.

(2) The overall data used in the study was from Feb-May 2018 to 2020. For comparisons between women and men’s submission, Feb-May 2019 and 2020 was used. For most countries, except China, social distancing measures and school closures did not happen until the middle of March (e.g. U.S.). Additionally, considering that for most of the manuscripts submitted in or before May, much of the work would have happened way before the world “shut down” due to the pandemic in most of the countries. Therefore, the time span used in the analysis is a little problematic. It will be useful to know what was the annual increase rate for submissions and reviews in Elsevier journals before the pandemic to rule out the possibility that the 30% increase in submissions between Feb-May 2019 and 2020 are in fact not due to normal annual increase.

Good point. Figure 1 [*included in the file version of the response letter] shows a smaller increase (11.7%) in submissions between 2018 and 2019, with a clear discontinuity in 2020 (+ 30.1 %). This corroborates the idea of looking at 2020 compared to the previous years as an exceptional period. We have added a reference to this discontinuity in the text. As regards the time window, by covering the whole period of March-May, we captured the effect of restrictions in EU and US, whereas with Feb 2020, we capture China and other Asian countries. Obviously, weekly instabilities could be present in the temporal pattern in specific places, but considering the whole period was instrumental to capture the big picture while considering country-effects. Note that from June on, anti-contagion measures were removed in many countries. So, extending the period to further months in the year would create other problems.

(3) For results in Table S4, the text in the manuscript says the dependent variable is the change in submissions for individual academics, while the supplemental materials indicate that the dependent variable is the number of submissions. The inconsistency needs to be adjusted.

The dependent variable in all models (both main text and SI) is the changes in the number of submissions. This is clearly indicated in the Table captions. Tab. S1 and S2 present descriptive statistics on the total number of submissions. This is also clearly indicated in the tables.

(4) On page 6: “in addition, we ….with more senior cohorts of women less penalized than younger scholars.” I am having hard time concluding this from the results in Table S4. I think it will be better if authors can explain a little bit more here.

We have improved the clarity of this part. 

(5) What is the percentage of “not title”, “doctor”, and “professor” category in the database? It will be useful to know in order to access the robustness of the analysis

We have reported detail by adding the percentages in the text.

(6) In all the analysis, how were those researchers entered “publishing” during the pandemic handled, considering they did not have publication histories prior to this? For the analysis on COVID-19 related submissions, this will be a bigger issue since there are a lot of people who did not publish on coronavirus related topics at all before they did so for COVID-19, which is particularly true of those non-biomedical COVID-19 publications.

Datasets (both global one and the COVID-19 ones) include all scientists who submitted at least on paper in the feb-may period of any of the three years. This could be an issue for covid-related papers indeed, since this was a minor topic before 2020. However, the keywords used by Elsevier to classify the COVID related papers, which are fully reported in the Methods section, include keywords that help to classify all coronaviruses, SARS (severe acute respiratory syndromes) and alikes before COVID 19 that can reflect the involvement of a scholar in this type of research before the explosion of the COVID pandemic. On the other hand, the alternative solution of including only authors who submitted in 2020 would have biased the results in the opposite direction, in the sense that it would have artificially inflated the 2020 increase in the number of submission (point 2 above). Note that our decision as to map the relative differences between men and women. Therefore, any increase regarding both groups (eventually driven by newcomers – those who submitted a manuscript in 2020 who did not in 2019 and 2018) is considered by the model intercept, which is not subject to any interpretation in the text. The relative difference between men and women is captured by the gender coefficient, which is the one we actually discuss in the text. These controls for the overall increase allowed us to directly test the hypothesis of women penalization (relatively to men) in terms of manuscript submissions.

(7) Based on the regression analysis results, the increase of submissions from female authors were smaller than that from male authors.

The referee is absolutely right. We have clarified this point when presenting Figure 1.

Reviewer #2: 

I find the paper very interesting, and I think the penalisation of (especially young) women is indisputable in all areas, including academia. A supporting analysis definitely deserves attention.

Concerning the methodology, I must say that I am not an expert in linear mixed models, but I have some general comments.

The first critical point concerns the data used. In a first analysis, to estimate the gender effects of the pandemic on submissions, the authors calculate the submission difference index for each author (dependent variables) as the difference between new submissions February-May 2020 minus the average number of submissions for the same months and for the same author in the periods 2018 and 2019. I wonder how plausible it is for the same author to submit an article in the same months in the previous two years. I think the presence of unobservable variables is not taken into much consideration. 

Other studies trying to estimate pre-post COVID used gender differences in the proportion of authors submitting to journals because they did not have any possibility to control for individuals in different time periods. Therefore, they neglected the in-out flow of different authors. We discussed this alternative (i.e., measuring the differences in the proportion of men and women authors in a comparable time period) but concluded that given that we had the possibility to reconstruct individual trajectories (the same individual in a three-year time series), our measurements were more precise. This required to keep a comparable Feb-May time window. First, this allowed us to capture seasonality effects. Secondly, if we would have considered other months, we should have been required to make assumptions on differences between numbers of submissions per authors per months. This would have super-complicated our analysis adding further layers of assumptions. Obviously, in these types of analysis, the size and potential effect of unobservable variables are an important problem, but this is often unavoidable with analysis on large-scale data on human behaviour. 

Moreover, February-May 2020 seems to me to be a very limited period, but above all too "early", to see the effects on scientific production, especially in the West. Only in Italy, one of the first countries to be affected, have there been closures since March. In this regard, the authors control for different "geographical" measures of prevention by including random effects for countries. 

You are right. Indeed, we included random effects on countries to take into account a number of unobserved variables, along with the grouping of our observation within distinct cultural and institutional environments. Also note that the period under consideration covers well the restriction that were implemented during the first COVID wave at least in East Asia (China, Korea, ...), Europe, and North America.

It is very interesting when they consider the scientific production on Covid-19 topics only, which surely has more concise working times, given the need of researchers to submit quickly. In that case, considering the time span February (maybe a bit early) - May seems to me more plausible.

You are right. We plan to extend our observation scales in future research. But, to reflect a point discussed also by Referee 1, adding more months in the summer 2020 could create other problems (given the easining of the anti-contagion measures in certain countries). We are negotiating with Elsevier to extend the observation periods to June 2021 in order to capture a richer time windows and more importantly measure the reviewer and editorial decisions. More than extending the observation period to map manuscript submissions, we believe that a temporal extension is required to understand the effect of peer review and editorial decisions. In any case, you should imagine how much it’s difficult to go back to manuscript submission systems, extract further data (which are incredibly dirty), clean them, enrich them with gender, seniority and other variables. We believe this paper is the first milestone of a line of research that we hope to develop even more in the future.

Furthermore, although it is clear that the focus of the paper is certainly not methodological, I think that a section (also in the supplementary materials) in which to motivate why linear mixed models have been considered and a minimum of formalisation is necessary. In addition, there is no test to verify the assumptions of the model (e.g. homogeneity, normality of error term, normality of random effect). Tests have probably been done, but this should at least be mentioned.

Thank you for your suggestion. We have added a section on data analysis where we explicitly mention that standard tests were performed to check the model assumptions (homogeneity, normality of random effect, etc.) for all models, if not explicitly otherwise mentioned in the text.

---

## [Decision Letter · Decision Letter 1]

14 Sep 2021

Gender gap in journal submissions and peer review during the first wave of the COVID-19 pandemic. A study on 2329 Elsevier journals

PONE-D-21-15330R1

Dear Dr. Squazzoni,

We’re pleased to inform you that your manuscript has been judged scientifically suitable for publication and will be formally accepted for publication once it meets all outstanding technical requirements.

Kind regards,

Alberto Baccini, Ph.D.

Academic Editor

PLOS ONE

Additional Editor Comments (optional):

Reviewers' comments:

Reviewer's Responses to Questions

**Comments to the Author**

1. If the authors have adequately addressed your comments raised in a previous round of review and you feel that this manuscript is now acceptable for publication, you may indicate that here to bypass the “Comments to the Author” section, enter your conflict of interest statement in the “Confidential to Editor” section, and submit your "Accept" recommendation.

Reviewer #1: All comments have been addressed

Reviewer #2: All comments have been addressed

2. Is the manuscript technically sound, and do the data support the conclusions?

Reviewer #1: Yes

Reviewer #2: (No Response)

3. Has the statistical analysis been performed appropriately and rigorously? 

Reviewer #1: Yes

Reviewer #2: (No Response)

4. Have the authors made all data underlying the findings in their manuscript fully available?

Reviewer #1: Yes

Reviewer #2: (No Response)

5. Is the manuscript presented in an intelligible fashion and written in standard English?

Reviewer #1: Yes

Reviewer #2: (No Response)

6. Review Comments to the Author

Reviewer #1: (No Response)

Reviewer #2: (No Response)

7. PLOS authors have the option to publish the peer review history of their article (what does this mean?). If published, this will include your full peer review and any attached files.

Reviewer #1: No

Reviewer #2: No

---

## [Editor Report · Acceptance letter]

22 Sep 2021

PONE-D-21-15330R1 

Gender gap in journal submissions and peer review during the first wave of the COVID-19 pandemic. A study on 2329 Elsevier journals 

Dear Dr. Squazzoni:

I'm pleased to inform you that your manuscript has been deemed suitable for publication in PLOS ONE. Congratulations! Your manuscript is now with our production department. 

Kind regards, 

on behalf of

Prof. Alberto Baccini 

Academic Editor

PLOS ONE